# Visual Methods to Assess Strain Fields in Armour Materials Subjected to Dynamic Deformation— A Review

**Chris L. Ellis** [1,2,]* and **Paul Hazell** [1]

[1]  School of Engineering and Information Technology, The University of New South Wales, Canberra, ACT 2600, Australia; p.hazell@adfa.edu.au

[2]  Land Division, Defence Science and Technology Group, 3207 Fishermans Bend, Australia

\*  Correspondence: Christopher.ellis1@adfa.edu.au; Tel.: +613-9686 8427



**Featured Application: A review of the measurement techniques of the dynamic high-strain rate response of ballistic protection materials experiencing high-speed impacts.**

**Abstract:** When impacted by a projectile, ballistic protection undergoes very large strain rates over very short periods of time. During these impact events, materials will undergo a very short region of elastic deformation, before undergoing significant plastic deformation. Due to the high levels of plastic deformation the samples undergo, strain gauges and other embedded sensors are often ineffective or become damaged before useful data can be obtained. Three-dimensional digital image correlation (3D DIC) is a non-invasive measurement method that uses two high-speed cameras, offset from each other by 15–45° to observe a speckle pattern on the sample material. As the material, and by extension the speckle pattern, deforms, the images taken throughout the deformation can be compared in sequence, to determine the motion and deformation of the sample. Recent advances in camera technology have allowed for frame rates in the hundreds of thousands of frames per-second, allowing for the measurement of very high-strain rate impact events. This paper will describe the premise of 3D DIC and provide a review of the current applications and research into high-speed impact testing using 3D DIC.

**Keywords:** armour; digital image correlation; experimental impact techniques

---

## 1. The Technique and Origin of Digital Image Correlation (DIC)

DIC is a method by which strain fields can be recorded across the surface of object as it deforms, without the need for strain gauges or other contact methods that may interfere with the deformation. This is accomplished by taking digital images of a random, high-contrast speckle pattern on the surface of the object. As the object deforms, the pattern on the surface also deforms and the changes in the pattern can be used to quantify the strain the object undergoes.

The method to carry out digital image correlation was first described in 1983 by Sutton et al. [1]. The authors described a method of taking digital images of an object prior to and post deformation and using the generated light intensity levels to convert the discrete data into a continuous form via a surface fit, shown in Figure 1. Due to the limits of computer technology in the 1980s, the overall images were broken down into small subsets for processing by the algorithm that the authors describe. The algorithm takes the subsets of the pattern pre-deformation and correlates them to the same subsets in the post-deformation surface. By using the algorithm that the authors describe, Sutton et al. were able to successfully measure beam deflections greater than 0.1 pixels [1].

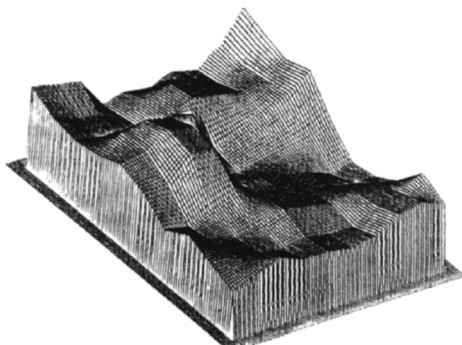

**Figure 1.** Example of an intensity pattern reconstructed using bilinear interpolation into a continuous form. Reproduced from Sutton et al. [1], with permission from Elsevier.

This method does require a couple of basic assumptions; (i) the pattern deforms exactly as the surface of the pattern, (ii) the light intensity patterns of the pre- and post-deformation surface have a unique one-to-one correspondence with the respective pre- and post-deformation surfaces [2]. In two-dimensional DIC, there is also a requirement that any out-of-plane displacement does not affect the in-plane deformation or displacement of the characteristic intensity surfaces and any derivatives of the out-of-plane displacement are significantly smaller than the derivatives of the in-plane displacements [2]. This is due to the deformation theory, described in [1], demonstrating that the observed intensity pattern is a two-dimensional projection of the object onto a plane [2]. Chu et al. [2] took the algorithm described in [1] and conducted a number of experiments to determine the accuracy of the system in three situations, (i) uniform translation, (ii) rigid body rotation, and (iii) uniform finite-strain test. The authors also compared the use of two different interpolation methods in the conversion of the discrete intensity map from the camera into continuous data; bi-linear and polynomial interpolation. The authors reported that the displacement components were within reasonable accuracy for both interpolation methods for the uniform translation experiments, while the rotation cases only agreed well (less than 4% error) for cases below 6 degrees of rotation. The last test the authors reported on found that the results for the uniform finite-strain test were very dependent on the image resolution, with large image resolution giving results between −18% to 8%, but lower resolutions giving up to −136% error [2].

In order to increase the accuracy of the DIC method, Bruck et al. [3] explored, using the Newton–Raphson method of partial differential corrections to increase the accuracy of displacement and gradient calculation. The Newton–Raphson method is based on the calculation of correction terms that improve the initial guesses of the DIC algorithm. By using the displacement of the subset centre, estimated to within one pixel, the algorithm is less likely to find a local minimum and more likely to find the absolute minimum. The authors also found that the Newton–Raphson method reduced computational time, when compared to the previously described coarse-fine search method. To evaluate the accuracy of the Newton–Raphson method in predicting displacement and gradient terms, the authors used three cases: (i) no deformation, (ii) translation only, and (iii) rotation only [3].

The first case of no deformation was undertaken, to determine the minimum error that could be expected in the system, and this is now part of the standard procedure of DIC [4]. The second experiment consisted of translations of 500 μm with a resolution of 87 pixels per cm (giving each translation at approximately 0.044 pixels) and the authors show good agreement between the actual and DIC results [3]. The third experiment the authors tried was rotation only, with ten rotations of 0.0157 radians, with two different subset sizes (50 × 50 and 20 × 20 pixels). In this case, the authors reported that larger subset size gave an order of magnitude improvement in the standard deviation for the predicted results [3].

As the technique advanced, methods were described to increase the accuracy of displacement and gradient calculations in digital image correlation. Sjödahl and Benckert [5] described a more

robust algorithm when it is subjected to decorrelation and displacement gradients. The idea of the new technique is to cross-correlate sub-images digitally from two similar, but relatively displaced, speckle patterns and apply a maximum search routine near the correlation peak, in order to obtain sub-pixel accuracy. While the authors concede that the formula the authors present is more time consuming, it can be performed in the spectral domain, through the use of a Fourier transform, and is significantly faster in the spectral domain. The authors demonstrate that to obtain a greater than 80% success rate of their algorithm, the sub-image size should be at least 10 times the speckle size [5].

By adding a second camera and turning the imaging system into stereo, out-of-plane displacements can be quantified, allowing for an accurate measurement of the three-dimensional strain of the object surface. Synnergren and Sjödahl [6] were the first to describe a three-dimensional DIC (3D DIC) technique and a calibration algorithm for such a system. The authors describe a physical model that relates points on an object with corresponding points in an image of that object. The shape of the object is measured by the cross-correlation of several sub-images from the two camera images, shown in Figure 2, through the algorithm described by Synnergren and Sjödahl. The two cameras are oriented so that the optical lenses are in parallel. This means that each camera can view the same image as the other camera, by moving the camera perpendicular to its optical access. The difference in the image fields is predominantly due to the shape of the object, and in order to reconstruct the shape of the object, the system must be calibrated. The calibration routine should be (i) easy to use, (ii) with a calibration target that is easy to make, and (iii) it should give accurate results. Synnergren and Sjödahl suggest a flat plate covered with a random dot pattern, that can move only in the z-direction (towards or away from the cameras), would satisfy the second requirement. The second calibration step would be to perform reference movements of the calibration plates. This is accomplished by moving the plate a known distance away from or towards the cameras and taking two sets of images, one before and one after the movement. The apparent movement due to the physical movement of the calibration plate is calculated for each camera. A least-squares optimization is then used to cross-correlate the apparent movement in one image, with the corresponding image from the other camera. To determine the accuracy of their 3D DIC method, Synnergren and Sjödahl performed the rigid body translation of a small model car and rotation of a suspended cylinder. For the rigid body motion, the authors reported that the main source of error in the rigid body motion was random error, with no significant influences originating from misalignment, distortion of the lenses or compensation for the shape of the object [6]. For the rotation of the suspended cylinder, the standard deviation of the rig the authors used was in the order of 0.15 milliradians. In comparison, the authors reported the algorithm and procedure accurately, predicting the rotation of the cylinder with between 0.13 to 0.17 milliradian standard deviation. The random errors were reported to be in the order of 1/100 of the pixel size used in the real experiments for the in-plane components and the out-of-plane errors were about six times lower [6].

As high-speed camera technology has advanced, 3D DIC has been extended to measure material deformation in areas where the strain rate can be quite high (in the order of $10^4$ /s). In order to perform high-speed (10,000+ frames per second, FPS) and ultra-high-speed (>1M frames per second) 3D DIC, several considerations need to be taken into account. The first of these factors is exposure time; limits in exposure time can cause blur, which lowers the algorithms' ability to attain sub-pixel accuracy [7]. The next factor is image quality, which is measured as high contrast and low noise. Very high levels of image contrast can be difficult to attain, as extremely short exposure times require high levels of scene illumination. A lack of contrast due to poor lighting can reduce the effective grayscale range of the image from 12-bits to 3- or 4-bits. Large amounts of lighting can be achieved with halogen lamps, but the heat from these light sources needs to be managed. Advances in LED lighting technology have allowed for bright lights that do not add significant heat to the sample. In terms of noise, this is usually related to the specific cameras, with some ultra-high-speed cameras having image intensifiers, which add noise. Noise of the system can be measured by the use of a speckled plate with images taken of the plate, while stationary and undergoing rigid body motion. Understanding how the cameras achieve

the high and ultra-high frame rates is also important. Some ultra-high-speed cameras utilize a rotating mirror to project the image onto multiple sensors. This technique can cause large misalignments in 3D DIC, due to the different optical pathways each image travels, and this makes cameras of this design unsuitable for 3D DIC. In addition, understanding the optical components, such as beam splitters, image intensifiers, and fibre optic bundles tying the intensifiers to the charge coupled device (CCD), and how the authors can cause a number of measurement uncertainties, such as variability blurring and non-radial distortions, can be very important [7]. While some have managed to develop a correction scheme to remove the complex distortion field in a 3D DIC setup, the variability of the displacement was ten times larger and the strain twenty times larger than a comparable high-speed complimentary metal-oxide semiconductor (CMOS) camera, even without the sample being moved [8]. Another major factor in using 3D DIC for high-strain rate events is ensuring that the paint and speckle pattern remains adhered to the deforming surface and are representative of the surface deformation [7].

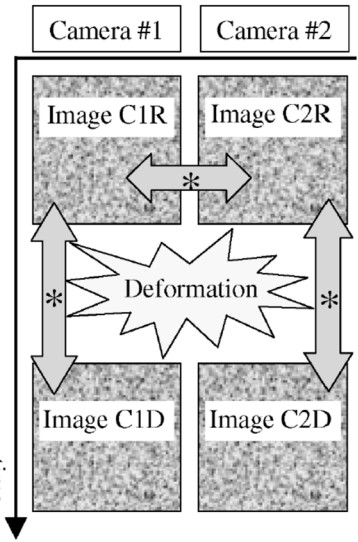

**Figure 2.** Time sequence, the arrows with * show which images are cross-correlated to find the 3D displacement field. Reproduced from Synnergren and Sjödahl [6], with permission from Elsevier.

With advances in digital X-ray camera technology, more research has been done on capturing multi-frame sequences with X-ray DIC. Jones et al. [9] performed a study comparing the errors in an optical 3D DIC system to an X-ray 3D DIC system, in situations where the optical system is known to operate poorly, such as imaging through distortions in the air caused by heat. To begin with, the authors performed a study, looking at methods of applying a speckle pattern with a denser material than the object of interest (in their case, an aluminium plate). The authors also examined the effect of plate thickness to speckle thickness to control contrast. The material the authors used for the X-ray speckle pattern was a thermally sprayed Tantalum powder with an aluminium mask to generate the speckle pattern. The authors used a thin adhesive layer to put the same speckle pattern for the optical cameras over the X-ray speckle pattern. While the two X-ray cameras (Varex 2520DX, Varex Imaging, Salt Lake City, Utah, USA) and the two optical cameras (PointGrey Grasshopper, FLIR Systems, Inc., Wilsonville, Oregon, USA) only ran at about five frames per second, the principles of X-ray speckle generation and the advantages shown are valid for higher frame rate X-ray photography. Both the optical and X-ray stereo systems were calibrated independently using the VIC-3D software (Correlated Solutions, Inc., Irmo, SC, USA). The authors conducted two experiments to compare the two systems, (i) pure translation of the speckle plate, and (ii) a stationary target, but with a hot plate in front of the target to cause optical distortions. For the first test, their results show that both the optical and X-ray systems were in good agreement, for both the in-plane and out-of-plane translations, and had similar noise floors. The results for the second experiment that the authors conducted showed significant

displacements with the optical system, even though the sample was not moving. In contrast, the X-ray system recorded only the random noise floor. The errors in the optical system recording displacement through heat distortions were 5–18 higher than the error of the X-ray system. The optical system also measured errors of 2–3 times higher in the strain measurements. This shows that X-ray 3D DIC could be a valuable tool in future to measure displacement and strain in regions, that are either not accessible via optical means, or are optically obscured [9].

One issue that can become pertinent in the use of 3D DIC is the resolving of different scales of phenomenon. Given the finite resolution of high and ultra-high-speed cameras, resolving the strain field for small scale localized deformation events can be at odds with determining the overall global strain field of the object being examined. An example of this is Patel and Peralta [10], where the authors examine the evolution of microbuckling in an UHMWPE laminate beam, using 2D DIC and the ARAMIS DIC software (GOM GmbH, Braunschweig, Germany), with a supported length of 152 mm. In comparison, the scale of the microbuckling is in the order of 600 μm [10]. Patel and Peralta [10] used a 100 μm thickness adhesive transfer tape and copper microparticles to generate a speckle pattern with a mean dot size of 15–20 μm and an inter-dot spacing of less than 10 μm. Applying a speckle pattern on a scale that can measure this microbuckling, but also being able to determine global strains can be challenging, especially when examined in three dimensions. Resolving these different scales of phenomenon requires the careful choice of field of view, an increased depth of field and a reliable speckle pattern that will still represent the underlying material deformation during large shear.

## 2. DIC Used in Impact Studies to Measure Target Deformation

Many armours are designed with an extremely hard strike face to defeat an incoming projectile. These materials are usually technical ceramics, such as aluminium oxide, silicon carbide, boron carbide and tungsten carbide [11]. These hard materials tend to spall during impact, leading to the requirement of a backing plate constructed of a softer material; usually a composite material constructed with polymer fibres, such as ultra-high-molecular-weight-polyethylene (UHMWPE), encased in an epoxy resin. Some other armours are constructed solely of the UHMWPE composite materials, such as combat helmets. The large deformations/strain rates that these composite plates undergo during a ballistic impact event are ideal applications for high- and ultra-high-speed 3D DIC.

In 2004, Schmidt et al. [12] reported on several projects being undertaken utilizing 3D DIC in impact testing, as a part of the NASA return to flight initiative. One of the crucial aspects of the NASA return to flight initiative is the extensive impact testing of foam projectiles on various orbiter subsections. As a precursor, their initial testing was focused on gas gun testing of coupons of the composite materials (6-inch reinforced carbon/carbon), being impacted by foam projectiles. The authors used a frame rate of 26,900 fps and were able to capture the entire initial deflection and recovery period and it agreed well with their computational calculations. With this success, the authors moved onto impacts on the orbiter wing leading edge. Due to very limited space within the wing leading edge, the authors used mirrors within the leading edge to lengthen the effective working distance from the cameras to the interior surface. The researchers determined that so long as the mirrors moved as a rigid body, with no local deformations, the authors concluded it would be the same as if there was no mirror. The authors showed that the noise floor of their setup had an accuracy of approximately 50 μm. With the mirrors, the authors were able to successfully capture the displacements from a local deformation at the impact point to a global buckling mode [12].

The US Army Research Laboratory (ARL) has had an interest in the use of 3D DIC in the impact of high velocity projectiles on armour materials for many years [13–17]. Early tests were conducted by striking a rigid polycarbonate and flexible poly(urethane urea) elastomer with spherical steel projectiles, with a diameter of 5.56 mm at a striking velocity of between 100 m/s and 200 m/s, which is below the ballistic limit of both materials [13]. By using Photron SA1 Cameras (Photron USA, Inc., San Diego, CA, USA), recording at 67,500 FPS and 256 × 256 pixel resolution, the authors were able to measure the strain field of the impacted specimens. The authors used a commercial image processing package,

ARAMIS (GOM GmbH, Braunschweig, Germany) to process the images, to give the full strain field. With this setup, the authors were able to produce a displacement accuracy to within ±25 μm, providing quantitative data that the authors could not obtain through other methods [13].

Another application of 3D DIC that has been examined is the back-face deformation of helmets to assess blunt force trauma. Modern helmets have become quite effective at defeating the penetration capabilities of ballistic threats, but often require large deformation to absorb the energy of the ballistic threat. If this back face deformation (BFD) is larger than the helmet standoff distance, this can impact a soldier's head, causing blunt force trauma [14]. Hisley et al. [14] developed an experimental methodology utilising 3D DIC at a frame rate of 50,000 FPS to produce a wide range of robust, accurate and repeatable time-dependent BFD data. As with the previous ARL study [13], the authors used the ARAMIS software (GOM GmbH, Braunschweig, Germany) to process the high-speed stereo photography of the BFD. A noise floor test demonstrated that their system had a noise floor of 0.6% of their maximum expected deflection [14]. The physical quantity that properly expresses the capacity to do work on tissue and cause blunt trauma from blunt impact is kinetic energy [18]. The ARAMIS software (GOM GmbH, Braunschweig, Germany) was able to provide instantaneous velocity of the back face of the helmet, allowing them to calculate the kinetic energy that the back face could impart onto the skull when it deforms to the helmet standoff distance [14].

Furthermore, 3D DIC has also been used to characterise the response of low velocity impacts on thin plates [19]. The purpose of the research conducted by Prentice et al. [19] was to assist in the design of explosive reactive armour (ERA). ERA consists of an explosive material sandwiched between two metal plates. When impacted, the shock waves generated can lead to the detonation of the confined explosive, pushing the plates apart and deflecting the projectile. By using chrome steel spheres fired at approximately 70 m/s at thin plates of copper and mild steel sheets, the authors were able to examine the early stages of penetration. The authors show that the early stages of penetration are typically dominated by ductile hole enlargement, where the target material is displaced laterally or backwards. Due to the significant plastic deformation in the plates, the authors reported that the paint coating with the speckle pattern was often damaged by forming radial cracks or spalling off between the radial cracks, highlighting the importance of choice of paint for high strain rate events. A sub-image size of 32 × 32 pixels and step size of 8 pixels gave a typical error that the authors show was approximately ±0.1 mm for in-plane deflections and ±0.05 mm for out-of-plane deflections. Given the low numbers of frames that some high-speed cameras can capture, the triggering mechanism is very important. Prentice et al. attempted to use two thin layers of electrically insulated brass on the point of impact as a trigger, but found that it had a significant effect on the plate response, highlighting the importance of non-invasive camera triggering mechanisms. The authors did report that while larger sub-image sizes gave smoother results, due to better signal-to-noise ratio and larger overlap between adjacent sub-images, it did reduce the spatial resolution, leading to under- or over-predicted displacements where the strain is non-uniform across the sub-image [19].

Vargas-Gonzalez et al. [16] from the US ARL used a pair of Photron SA2 cameras (Photron USA, Inc., San Diego, CA, USA) to obtain 3D DIC measurements of the back face displacement of hybridized and thermoplastic laminates when impacted with various projectiles. The authors used the ARAMIS software (GOM GmbH, Braunschweig, Germany) for the processing of the high-speed imagery to obtain the deflection fields. This particular study was aimed at the reduction of back face deformation, through the use of hybridized materials or the clever use of architecture and fibre orientation. A baseline of laminates with an orientation of [0°/90°] fibres of UHMWPE (specifically SpectraShield II 3130 and Dyneema) was used, with new versions of laminates laid up in a quasi-isotropic manner, with every two plies rotating clockwise 22.5° (referred to as multi-orientation). Vargas-Gonzalez et al. did ensure that all of the panels had the same areal density (10.74 kg/m²). The panels were impacted with a 5.56 mm diameter 440C steel ball bearing, at a velocity of 405.7 ± 6.7 m/s and 9 mm diameter, and a 124-grain full metal jacket round at a velocity of 473.13 ± 3.6 m/s. Using the 3D DIC methodology, the authors show that the back-face deformation was reduced significantly with the multi-orientation

panels, but it also reduced the panel impact resistance. Overall, the authors show that a 50%/50% combination of multi-orientation laminates on the strike face and conventional [0°/90°] laminates on the back face provided the best performance, with both lowest back face deformation and highest projectile resistance. Vargas-Gonzalez et al. came to the conclusion that there is a trade-off between stiffness and penetration resistance [16].

Vargas-Gonzalez et al. [17] continued their research on the relationship between ballistic and structural properties of lightweight thermoplastic, unidirectional composite laminates. The experiments the authors conducted consisted of impacting Dyneema HB26 and Spectra Shield II SR-3136 laminates with 75% of the mass of the panel (strike face) of [0°/90°] orientated fibres and the rear 25% laid up in a quasi-isotropic layup, where every two succeeding plies are rotated 22.5° (known as ARL X hybrid). As a baseline, the authors used Kevlar KM2 style 705 PVB phenolic woven aramid composites and Spectra Shield and Dyneema panels with [0°/90°] orientation layups (referred to as monolithic [0°/90°] panels). All of the panels were constructed to have the same areal density. The impact tests were conducted with 0.22 caliber fragment simulating projectiles (FSP) to determine V50 and 9 mm, 124 grain full metal jacket projectiles to measure back face deformation and for the 3D DIC studies. To capture the high-speed imagery, the authors used a pair of Photron SA5 cameras (Photron USA, Inc., San Diego, CA, USA) with a frame rate of between 40,000 and 50,000 FPS and used the ARAMIS software (GOM GmbH, Braunschweig, Germany) to process the imagery into deflection and strain fields. By doing this, the authors were able to calculate the energy on the back face of the panel during impact events using the instantaneous velocity of the panel back face, the effective area of the delamination and the areal density of the material participating in the impact event [17]. Their 3D DIC results demonstrated that the ARL X hybrid panels have a significantly higher interaction area than the monolithic [0°/90°] panels, due to the stress being spread out in a more circular pattern, due to the increased numbers of angles of fibre directions. Comparatively, the monolithic [0°/90°] panels exhibit delamination in the 0° and 90° directions. The higher interaction area in the ARL X hybrid panels allow them to dissipate more energy into the panel over time, as captured by the 3D DIC. The plots of the ratio of available energy to interaction area demonstrate that the energy in the ARL X hybrid panels is more diffuse, reducing the back face deformation [17].

Advances in X-ray technology have allowed for 3D DIC to be extended to tracking the motion within objects, by seeding them with particles of significantly higher density. One example of this is a study conducted by Rae et al. [20], where the authors examined the displacement ahead of a spherical projectile being fired at a mock (non-reactive) motor casing. The mock motor the authors used was made of 6061 aluminium tubing, a Sylgard 184 insulation layer and a mock explosive (PBS 9501) on the inside. The authors describe two versions of the mock motor casing, one with a central round 'burn' hole and one which is solid. The projectile the authors used was an 8 mm diameter hardened steel ball bearing, fired with a velocity of 490 m/s. The X-rays were developed on film, and then scanned in as 8-bit greyscale digital images. To obtain speckles in the X-ray images, the mock explosive was seeded with lead particles, which gave a significantly larger contrast than the surrounding particles. The lead particles the authors used were sized at <1.5 mm in diameter. The authors show that using only the smaller particles gave crisp, but unacceptable large speckles. In comparison, a non-contiguous layer of lead granules, together with about the same mass of 1.5 mm diameter particles, gave more acceptable speckles for DIC purposes. The X-ray impact recording was triggered by the steel ball shorting out a thin, flexible Mylar/copper printed circuit tape, bonded to the outside of the aluminium tube. The authors modified the delay time from the trigger for a number of repeats of the same experiment, to give a sequence of images throughout the impact event. Due to a lack of easy optimization of the speckle size and contrast, the authors decided to compare three different correlation techniques on the X-ray images. The three correlation methods were: (i) the Vic-2D (Correlated Solutions, Inc., Irmo, SC, USA) commercial software package, (ii) making use of a fast Fourier transform to change the data to the frequency domain, with sophisticated sub-pixel image shifting to optimize the overlap between sub-images and (iii) also making use of a fast Fourier transform to perform the correlation, but utilizing

a simple bi-cubic fit to find the maximum correlation peak, to find the average displacement between sub-images [20]. Rae et al. found that all three correlation methods produced some valid data from the sub-optimal speckle patterns around the projectile in the case of the solid sample, but method (ii) produced poor results around the impact region for the case with the 'burn' hole [20].

The aim to develop better materials for ballistic helmets is predominantly focused on weight reduction with the maintenance of ballistic protection performance. This reduction in weight often comes with thinner materials, which can lead to greater back face deformation, and potentially increase behind helmet blunt force trauma [21]. This leads to future helmet design needing to balance weight, ballistic performance and back face deflection. Freitas et al. [21] examined a set of seventeen different composite materials, measuring ballistic limit and dynamic back face deflection through the use of 3D DIC. The projectiles that the authors used were a 9 mm full metal jacket, 0.30 caliber fragment simulating projectile and a 0.22 caliber fragment simulating projectile, which form what is considered to be a traditional set of small arms or fragmentation threats. There were a number of characteristics that the authors obtained from the DIC measurements. These were: (i) dynamic deflection time history, (ii) back face velocity time history, (iii) strain time history, (iv) spatial distributions, and (v) back face shape during loading and unloading. The authors used a pair of Vision Research Phantom v710 cameras (Vision Research Inc., Wayne, NJ, USA), running at 30,000 FPS and the ARAMIS software (GOM GmbH, Braunschweig, Germany) for image processing. Freitas et al. reported on the full-time history of the back face deformation for four target materials; Dyneema HB80, Spectra Shield 3130, Spectra Shield 3124 and Tensylon HTBD. The authors reported that the Dyneema and two Spectra Shield targets respond to ballistic impact over a longer time frame in an elastic mode similar to that of fabric armour. The authors describe this response as taking the form of a single longitudinal displacement wave that propagates relatively slowly towards the boundaries of the target. In contrast, the Tensylon target exhibited a more complex waveform over an equivalent time frame. This more complex waveform is attributed to localized failure in the region of the impact site dampens the deflection wave. This has the effect of reducing the back-face deformation, but also lowering the V50 velocity as well. The deformation in the Tensylon back face is localized to the impact region and has a significantly faster transmission of the deflection wave form, which indicates a stiffer material [21].

The use of 3D DIC is not just isolated to the measurement of purely composite panels. O'Masta et al. [22] examined the dynamic response of a panel composed of triangular prisms made from alumina ceramic, encased in aluminium with an outer casing of UHMWPE fibre reinforced laminates, shown in Figure 3. Schematic diagram of the target used in O'Masta et al. [22] To measure the back face deflection, the authors removed a section of the UHMWPE fibre reinforced laminate $100 \times 100$ mm and covered this region of bare aluminium in a speckle pattern, to allow for 3D DIC. The authors used a pair of SA-X2 Photron cameras (Photron USA, Inc., San Diego, CA, USA) to obtain the high-speed imagery and the ARAMIS software (GOM GmbH, Braunschweig, Germany) to process the images. For the DIC testing, the authors used a powder gun, which was capable of launching the 12.7 mm diameter spherical projectile at speeds of up to 2.3 km/s. For the prism base impact, the authors show that when impacted by the projectile at 1.71 km/s, a small bulge formed 4 µs from impact, but that the bulge was longer in the longitudinal direction (along the prism long axis) than the transverse direction. Around the time of peak deflection, the dimensions of the bulge and its velocity increased, and a y-oriented tear in the face sheet started to develop below the node of the centrally impacted cell. The authors measured the peak deflection at around 6 mm before ejecta exited the tear. Using the high-speed imagery, the authors were able to determine the plume of debris had a blunt front and was about 44 mm wide in the y-direction, consistent with the dimension of the face sheet tear opening. The authors were able to use the DIC to determine the velocity at the most deflected location, reaching 0.30 km/s for the 1.71 km/s projectile speed and 0.45 km/s for the 2.29 km/s projectile speed. For apex impacts, the authors reported that a pair of bulges occurred on either side of the base of the apex of the ceramic prisms. By 10 µs, the peaks had merged into a single 20 mm wide transverse front. From this bulge, two y-oriented cracks formed, allowing a portion of the rear face sheet under

the impacted prism to be torn away. The authors measured the residual velocity and found it to be approximately double that of the prism base impact [22].

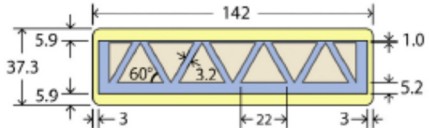

**Figure 3.** Schematic diagram of the target used in O'Masta et al. [22] Reproduced from O'Masta et al. [22], with permission from Elsevier.

Further research into the deformation of UHMWPE composites with the aim of preventing behind helmet blunt force trauma was conducted by Zhang et al. [23] The authors undertook this study due to UHMWPE fibres exhibiting large strains to arrest projectiles and a need to characterise the ballistic response of different UHMWPE fibres. The authors recorded the BFD with two Photron SA5 cameras (Photron USA, Inc., San Diego, CA, USA) recording at between 40,000 and 50,000 FPS and using the ARAMIS software (GOM GmbH, Braunschweig, Germany) to perform the DIC image processing. The authors examined several different UHMWPE fibres, Dyneema HB25 and HB80, and Spectra Shield II SR-3136 fibres, and used a woven aramid K705 laminate panel as a baseline. All of the panels were made to have the same areal density, 7.8 kg/m$^2$. To determine the V50 of the panels, the authors used a 0.22 caliber fragment simulating projectile and used a 12.7 mm steel sphere projectile to determine BFD (with a lower velocity to prevent perforation). Their results show that the ARL X hybrid panels constructed with Dyneema HB25 fibres exhibited a 10% drop in resistance to penetration, but a greater than 45% decrease in BFD. In comparison, their experiments using the Spectra Shield II SR-3136 fibres exhibited no decrease in resistance to penetration, but still retained a 40% decrease in BFD. Using the 3D DIC, the authors show that the peak BFD occurs after the initial impact, and is larger than the final BFD, indicating some elastic recovery. The ARL X hybrid panels exhibited larger lateral spread of deformation and delamination, but smaller out-of-plane deformation [23].

Ćwik et al. [24] used multiple high-speed cameras to perform 3D DIC simultaneously, on both the front and back of composite panels constructed of [0°/90°] cross ply layups of Dyneema HB26 and Spectra Shield 3124 UHMWPE fibres. The authors used a combination of pairs of cameras including the Phantom v711, Phantom v12 and Phantom v16 (Vision Research Inc., Wayne, NJ, USA), running at between 21,000 to 82,000 FPS. To ensure that the photography was synchronized, all of the cameras ran off a common trigger and common time reference [24]. The authors extended the speckle pattern onto the rig, holding the samples to identify if it moved during the experiment. The projectiles used to test the samples were 20 mm diameter steel and copper fragment simulating projectiles fired at a velocity of up to 1.2 km/s. The initial tests carried out were V50 tests and found that the ballistic limit for the Dyneema HB26 composite panel was around 700 m/s for both projectiles and about 600 m/s for the Spectra Shield 3124 impacted with the steel projectile and 700 m/s for the copper projectile. The 3D DIC on the front face of the Spectra Shield 3124 revealed that a certain part of the front face material experienced an inelastic deformation, but only experienced less than 1% tensile strain during the impact event, regardless of impact velocity. It also revealed that a bright flash, circular in shape, occurred during the instance of projectile-panel impact. This flash changed to a peanut shape as the projectile began to penetrate the panel. After the flash had dissipated, melted material was ejected from the sides of the projectile chiselled nose. After the impact, the 3D DIC revealed that the panels underwent in-plane motion before out-of-plane motion, due to the sonic velocity along the Dyneema fibres being significantly higher than the transverse velocity through the panel. This in-plane motion takes the form of wrinkles propagating towards the edge of the panel. Overall, the authors show that the Dyneema HB26 undergoes smaller out-of-plane displacements than the Spectra Shield 3124. In addition, the in-plane area of deformation for the Spectra Shield 3124 panels is significantly smaller than the Dyneema, likely due to a higher shear strength and stiffness in the Dyneema. As for 3D DIC conducted on the back face of the panels, the authors demonstrated that the primary and secondary

yarns at the back of the panel did not experience more than 2% throughout most of the measured length, regardless of strike velocity. The primary yarns experienced the greatest strain throughout their lengths immediately after the flash at the front face vanished [24].

Given the recent increased threat of laser based systems, Kjawinska et al. [25] conducted a study using 3D DIC of the interaction between a high powered laser and carbon and glass fibre structural materials. For the 3D DIC, the authors used a pair of Optronis CR3000 cameras (Optronis, Kehl, Germany) running at 500 FPS and the VIC-3D software (Correlated Solutions, Inc., Irmo, SC, USA) to process the images and a single FLIR SC 7500 camera (FLIR Systems, Inc., Wilsonville, Oregon, USA) for infrared imagery. This study tested two different sample geometries, a thin carbon or glass laminate with 1.5 mm of foam core and a thick carbon or glass laminate with 5 mm foam core. The laser was a Nd:YAG type with a wavelength of 1.06 μm, 10 Hz impulse frequency and an impulse duration of 2 ms. Each single impulse had 5 J of energy and was focused onto the target by a lens that gave an impact circle of 2 mm diameter. Each sample was impacted by 10 pulses. The results of their 3D DIC indicated that the internal pressure increased significantly, due to thermal degradation of the matrix and foam. The authors reported that the foam acted as an insulating material for both thermal transition and penetration of the laser light, significantly increasing the resistance of the structure to laser damage [25].

With advances in camera resolution and frame rate, obtaining an optimized speckle pattern becomes more important. Jannotti and Schuster [26] designed a rig to hold a permanent marker in a custom fixture with a 3 axis milling machine providing accurate location control. Their custom fixture allowed for precise creation of a pre-generated speckle pattern, with specified speckle size. This control allows for the production of speckle patterns that give both 3 pixels per speckle and 50/50 black to white coverage. Using this rig, the authors were able to make patterns with about 3% spatial resolution (about 50 μm), sized from 0.2 mm to 4 mm and coverage areas of 10 mm × 10 mm to 200 mm × 200 mm. Two Specialised Imaging Kiranas cameras, recording at 2 million FPS, were used to obtain the imagery for the 3D DIC. To test the speckle patterns with 3D DIC, the authors impacted 6.35 mm thick aluminium plates with, 12.7 mm diameter copper spheres at 1.2 km/s. The authors recorded a bulge of 18 mm in diameter and displaced nearly 9 mm prior to failure of the plate (which occurred in less than 15 μs). Through the 3D DIC image processing, the authors were able to calculate the von Mises, radial tensile and radial compressive strains throughout the impact of the projectile into the plate [26].

Janotti [27] continued their work in 2019, by evaluating the underlying elements of uncertainty that affect collected data, during high-strain rate mechanical testing [27]. To test this, 3D DIC was conducted on a 6.35 mm thick aluminium disk, being impacted by a 6.25 mm diameter steel sphere with a velocity of approximately 500 m/s. The authors used a pair of Shimadzu HPV-X2 cameras (Shimadzu, Kyoto, Japan) operating at up to 10 million FPS. The DIC system was tested prior to the ballistic testing, investigating warm-up error, and displacement error. During the initial warmup (30–60 min), the cameras exhibited a sharp increase in temperature, leading to in-plane and out-of-plane errors of up to 60 μm and 30 μ, respectively. By 60 min of warmup, the error had decreased to 5 to 10 μm and had decreased to 1 μm by the 120-minute mark. Once the cameras had fully warmed up, Janotti examined the measurement error of the system, by displacing speckled target distances of 1–13 mm in increment of out-of-plane steps of 0.1, 0.2, 0.5 and 1.0 mm steps. The authors show that the error was in the range of 5–11 μm (0.5–6% of the measured step size, respectively). As the step size decreased, the percent error increased as the precision limit of the DIC system was approached. The ballistic tests conducted on the 6.35 mm thick aluminium disk showed a peak surface displacement and velocity of 3 mm and 40 m/s when impacted by the steel sphere at 536 m/s, with an average interframe displacement of 120 μm. The error for this range of displacement was taken by Janotti as being between 0.5% and 6% [27].

Chocron et al. [28] used 3D DIC to examine ballistic impact loading on carbon fibre materials, with a thickness of 6.35 mm and 25.4 mm. The authors used two different carbon fibre layup architectures,

one being 2D layups laminated into a single panel and the other with a complex 3D panel. The 2D carbon fibre panel was laid up in a [0°/45°/−45°/90°] stacking sequence with 2 stacks used for the nominal thickness of 6.35 mm, and 8 stacks were used and for the 25.4 mm thickness. The 3D panels had a complex weave, with differing architecture between the 6.35 mm and 25.4 mm thickness panel. Both the 2D and 3D panels, with a nominal thickness of 6.35 mm, had an areal density of 0.97 kg/m$^2$, and the 2D and 3D panels with a nominal thickness of 25.4 mm had an areal density of 3.9 kg/m$^2$. The authors used a mirror behind the target to prevent damage to the two Phantom v7.11 cameras (Vision Research Inc., Wayne, NJ, USA) that were used to measure the BFD. The authors also used a Shimadzu HPV-X2 camera (Shimadzu, Kyoto, Japan), placed laterally to the target. This camera was used to record the evolution of the impact triangle and to measure the residual velocity of the projectile, if it perforated the target. The projectiles used were a 0.30 caliber fragment simulating projectile, fired at between 265 m/s and 1050 m/s to determine the V50 of the composite laminates. The authors show that the 25.4 mm carbon fibre panels impacted at 362 m/s showed a BFD of 0.45 mm at 38 μs after impact. In comparison, the maximum deflection was 4.5 mm for an impact speed of 848 m/s. The maximum displacement results shown for the 3D DIC and the Shimadzu cameras were very similar (4.5 mm and 5 mm, respectively), with the small difference being attributed to frame rate difference or measurement accuracy. The 25.4 mm thick 2D panel exhibited a V50 of 1050 ± 106 m/s, with the 6.35 mm thick 2D panel exhibiting a V50 of 282 ± 73 m/s. The 3D composite panels exhibited similar V50 to the 2D panels with the same thickness, but the authors did exhibit about 70% smaller damage areas, leading to greater damage tolerance [28].

Sarvestani et al. [29] examined a number of novel multi-layered architectured ceramics, laminated together to enhance damage tolerance. Each ceramic tile was made from 263 borosilicate ceramic with 96–99.8% theoretical density, 3875 kg/m$^3$ density and a thickness of 0.635 mm. The authors created three different designs for the panels. The designs consisted of (i) hexagonal tiles with three different characteristic lengths ($L_1$), (ii) parallel cut tiles with two different spacing lengths ($L_2$) to develop multilayered cross ply (4 layer) and bouligand (8 layer) ceramic panels and (iii) a shell like design with a combination of hexagonal and plain configuration. All of the tiles were bonded together using Surlyn adhesive, as it provides an elastomeric response with a strain stiffening mechanism at the interfaces. This delays localization fracture and distributes the deformation throughout the material. The authors performed both quasi-static and impact tests on the architectured ceramics, with the side opposite the load or strike face being speckled to allow for DIC. For the quasi-static tests, the authors used two five-megapixel cameras (FLIR Grasshopper3, Systems, Inc., Wilsonville, Oregon, USA), illuminated with a high intensity LED light. The imaging for the impact testing was accomplished using two one-megapixel cameras operating at 20,000 FPS, and this imagery was processed to obtain the deformation and strain fields. Both the quasi-static and impact tests were performed three times in different locations near the centre of the tiles, ensuring the hit locations were the same between tile samples. The impact testing was conducted with a 14 J low velocity impact (2.1 m/s impact velocity). The authors show that smaller hexagonal tiles ($L_1$ = 2.5 and 5 mm) with lower stiffness increased the multi-hit capability, absorbing more energy in the second and third hit than any of the other arrangements. However, decreasing tile size does lead to larger delamination and greater deflection [29].

Domun et al. [30] examined, using 3D DIC, the full field deformation and major in-plane strains of glass fibre reinforced polymer laminates with a nano-modified epoxy matrix. The 3D DIC was conducted with two Phantom Miro M/R/LC310 high-speed cameras (Vision Research Inc., Wayne, NJ, USA) at a frame rate of 39,000 FPS and the imagery was processed using the ARAMIS commercial software. The epoxy matrices were modified by including graphene nanoplatelets, multi-walled carbon nanotubes, multi-walled boron nitride nanotubes or hexagonal boron nitride particles. The composite panels were made of plys in an arrangement of [(45°/90°/−45°/0°)$_3$ / (0°/45°/90°/−45°)$_3$]. Two projectile speeds were undertaken, a high energy impact velocity of 134.3 ± 1.7 m/s and a low energy impact test of 76 ± 1 m/s. The projectiles the authors used were cylindrical, with a hemispherical nose, a diameter

of 24.9 ± 0.1 mm, and made of an aluminium alloy. The high-speed tests were undertaken to measure the specific energy absorption of the panels, by looking at the residual energy of the projectiles after perforation. The authors undertook the low-speed tests to measure the full field deformation and major in-plane strains for non-penetrating impacts. The authors show that the durations of the impact were identical for all of the low-speed impacts, with the neat epoxy (no nano-modification) having the most compliance and showing the greatest strains. The neat epoxy also retained the largest residual strain at the end of unloading, implying the greatest permanent damage. For the high-speed impacts, the authors show that all of the nano-modified panels recorded exit velocities lower than the neat epoxy panel. The panel with a mix of epoxy, boron nanotubes and graphene nano-platelets performed the best, with a reduction in exit velocity of 89.1% [30].

Hild et al. [31] used a single Shimadzu HPV-X camera (Shimadzu, Kyoto, Japan) running at 5M and 10M FPS to perform 2D DIC on two dynamic Brazillian test performed in split Hopkinson pressure bars. The authors reported on five different kinematic fields; displacement, velocity, acceleration, strain and strain rate fields. Brazillian tests enable the tensile strength of brittle materials such as concrete and rocks to be assessed. The authors used a 72 mm diameter and 10 mm thick disk made of ductal concrete, loaded to failure under impact, for each test. The speed of the input bar of the Hopkinson bar was about 6 m/s. Their first set of experiments used 5M FPS and demonstrated the reduction in displacement and strain uncertainty by using spatiotemporal DIC, rather than instantaneous DIC, was rather small, ~4 times, but reduced significantly more for velocity and strain rate (~$4^3$) and even more for the acceleration (~$4^5$). The authors show that the second set of experiments, utilizing the cameras at 10M FPS, produced results consistent with the first experiment [31].

Liu et al. [32] published their initial findings of a set of experiments comparing carbon fibre (CF)/epoxy and CF/poly(ether-ether keytone) (PEEK), in both drop rig tests and gas gun tests. The authors utilized 3D DIC in the gas gun tests, to determine BFD during the impact event. The high-speed photography was taken with two Phantom Miro M/R/LC310 cameras (Vision Research Inc., Wayne, NJ, USA), running at 29,000 FPS. The low-speed tests utilized a 12.7 mm diameter instrumented steel impactor, with a mass of 5.37 kg. The authors utilized a catching mechanism to prevent a second strike after rebound. The projectile for the high velocity tests was a hemispherical 12.7 mm diameter aluminium alloy projectile. The authors plotted the loads experienced during the drop tests, with each sample experiencing an initial increase in load due to the elastic response of the material, then a decrease in load, indicating initial damage, before a rise to maximum load. After the initial damage has occurred, the authors recorded the propagation of delamination, and matrix cracking from the initial damage. After the maximum load, the impactor rebounds, leading to a smooth decrease in the load. The authors reported that the load at the initial damage was relatively independent of the impact energy level. The CF/PEEK composites showed higher peak load than the CF/epoxy for all impact energy levels. The CF/PEEK also demonstrated less damage than the CF/epoxy specimens. In this paper, the authors only gave the out-of-plane displacement (2.13 mm) for the CF/epoxy panel, impacted by a projectile with a velocity of 46 m/s [32].

Liu et al. [33] continued their work into the response of carbon fibre reinforced polymers to impact, by using a gas gun to launch both hard and soft projectiles at CFRP panels and recording the response with 3D DIC. Two high-speed cameras, Phantom Miro M/R/LC310 (Vision Research Inc., Wayne, NJ, USA), running at 39,000 FPS were used to capture the images of the BFD during the impact, with the ARAMIS software (GOM GmbH, Braunschweig, Germany) being used for 3D DIC image processing. The extension of impacts to soft projectiles was done to simulate an event like bird strike, where the projectile is highly deformable. Wilbeck and Rand suggested that a gelatin projectile, with a length to width ratio of 2 and a density of 1.03 g/cm$^3$, would be suitable as a bird stimulant for small scale laboratory tests [34]. The projectiles Liu et al. [33] used were a gelatin projectile with a mass of 20 g, a diameter of 24.9 mm and a length of 37 mm and a hemispherical nosed T7075—T6 aluminium projectile, with a diameter of 24.9 mm and a length of 31.4 mm. The projectile speeds the authors used ranged from 42 m/s to 100 m/s. Their DIC measurements indicated that the CF/PEEK exhibited a

maximum out-of-plane BFD of approximately 3.7 mm, when impacted by the soft projectile at an impact velocity of 100 m/s. For the same impact velocity and projectile, the CF/epoxy combination exhibited a 10% greater maximum out-of-plane deformation and a slower rebound sequence. The authors attributed this increased out-of-plane deformation and slower rebound sequence to significant damage found on the CF/epoxy panel after impact. In comparison, the CF/PEEK panels exhibited no visible damage for an impact of the soft projectile at 100 m/s. For hard projectile impacts, a projectile velocity of 50 m/s was sufficient to perforate the CF/PEEK panel. In comparison, the CF/epoxy panel was also perforated at a projectile velocity of 50 m/s, but the area of damage on the panel was approximately three times larger. In addition, the damage area did not increase in size for projectile velocities above 50 m/s for either panel [34].

Wen et al. [35] conducted an investigation into the impact of a 9 mm projectile impact on a combat helmet worn by a human head surrogate. The authors used 3D DIC to measure the front face deformation, while the human head surrogate had sections filled with bio-simulant clay (specifically Roma Plastilina No.1 oil-based modelling clay). The combat helmet the authors tested was a composite, consisting of aramid-based fibres with a phenolic-based resin matrix. The projectile was a full metal jacket 9 mm pistol round, with an initial velocity of 335 m/s. The 3D DIC images were taken at 10,000 FPS and a resolution of 1024 × 800 pixels. No information was given about the software package used to process the images or what cameras were used. The authors show that the front face experienced a maximum dynamic deformation of 21.4 mm and affected an oval domain of approximately 62 mm short axis. The final deformation was approximately 12 mm and affected a circular area of approximately 51 mm. No indentation was seen in the clay head form, indicating that there was no perforation and that the helmet shell deformation does not exceed the standoff distance of 20mm. Using the 3D DIC, the authors also found that the maximum velocity of the front face increases sharply to a maximum of 21 m/s in the first 1.2 ms after impact [35].

Seidl et al. [36] conducted a study to investigate the BFD of a 7.62x39 mm oblique hit on panels made from an Aramid fibre composite used in combat helmets. Initially, the authors used perpendicular hits on the composite panels with a 9 mm lead core projectile, to calibrate the 3D DIC system. Seidl et al. reported that the authors obtained the panels directly from a helmet manufacturer and did not receive information regarding the fibre directions of the panels. The authors did report that the panels had 23 plies and a thickness of 8.5 ± 0.5 mm. The authors used a combination of a Phantom v311 and Phantom v310 cameras (Vision Research Inc., Wayne, NJ, USA), running at 15,000 FPS, a resolution of 512 × 384 pixels and 4 μs exposure time to capture the high-speed images of the back face. No information was provided regarding the 3D DIC processing software the authors used. While the initial 9 mm projectile shots were used to calibrate the 3D DIC system, the authors also conducted a small study on multiple hits on the panels (without overlapping areas of effect), looking to determine if a difference in the BFD occurs and if the 3D DIC was capable of capturing it. Their results show that there was no difference in the BFD between the first and second shot on the same plate. However, there was less oscillation on the target plate on the second shot if there is pre-damage on the plate. The impacts with the 7.62 × 39 mm projectiles were conducted with a muzzle velocity of between 500 and 720 m/s and at an angle of obliquity of 70°. The authors investigated two difference sized panels (200 × 300 × 8.5 mm and 100 × 300 × 8.5 mm) for these tests, to determine if the distance to the boundaries affected the panel response to impact. The authors show that the region of plastic deformation was smaller in the smaller panel and the larger panel exhibited greater dynamic elastic deformation. The larger panels did exhibit larger displacement in the z-direction, showing that the data from the smaller and larger panels, for the same projectile and muzzle velocity, were not comparable [36].

As this section shows, 3D DIC is rapidly becoming a very effective tool for determining both the front and back face deflection during ballistic impacts on the flat and curved panels of a variety of materials.

### 3. 3D DIC in Blast and Shock Loading

One of the earliest studies utilizing 3D DIC to measure the dynamic deflection of plates loaded through explosive blasts was a study by Tiwari et al. [37] This study examined the response of aluminium 6061 sheets (356 × 406 × 1.6 mm) to sub-surface detonations with a fixed stand-off distance. Two depths were used for the explosive, 7.6 mm and 25.4 mm, with a stand-off distance of 28.7 mm. Tiwari et al. used a pair of Phantom V7.1 high-speed cameras (Vision Research Inc., Wayne, NJ, USA) to capture the images. The authors used a field of view of 75 × 75 mm and a frame rate of 61,538 FPS for the shallow depth detonations and a field of view of 150 × 150 mm and 26,143 FPS for the deeper detonations. Their results demonstrate that a shallow explosion causes a vertical deformation in the plate which is twice as large as the deep explosion, with similar trends in vertical velocity and vertical acceleration. The shallow plate experienced a maximum velocity of 230 m/s during the early stages of the blast loading. The 3D DIC also provided information regarding the strain rates experienced by the aluminium plates. The shallow detonation caused higher strain rates, but these were sustained over a shorter period of time than the deeper detonations [37].

Arora et al. [38] conducted a study looking at the response of sandwich panels (back and front face sheets of glass-fibre reinforced polymer with a styrene acrylonitrile foam core) to explosive blasts in free air. The face sheets consisted of two plies of [0°/90°/±45°] e-glass each and the foam core had thickness of 40 mm. The explosive used in that study was 30 kgs of C4 with standoff distances of 8 m and 14 m. The intent of the 8 m standoff was to cause significant damage to the sandwich panels, while the 14 m standoff was to impart a pressure of ~200 kPa to the sandwich panel and take them to their elastic limit, with no visible blast damage effects on the front face. The 3D DIC images were taken with a pair of Photron SA3 cameras (Photron USA, Inc., San Diego, CA, USA) sampling at 2000 FPS with a resolution of 1024 × 1024 pixels. The authors used the ARAMIS software (GOM GmbH, Braunschweig, Germany) for DIC image processing. To verify the 3D DIC results, the authors used a laser gauge as a secondary displacement measurement device, focused on the centre of the panel. With the larger standoff distance, the results show that there was a uniform and symmetrical response of the panel, up until the maximum out-of-plane displacement of 63 mm. Both the laser gauge and DIC data correlated well, up until the point of maximum deflection. After this point, vibrations transmit through the isolation mounts to the camera arrangement, changing their position relative to their original positions. There was no evidence of damage to the panel shown by their DIC analysis in terms of magnitude or distribution of strain. In comparison, the closer standoff distance resulted in significant damage to the panels. Initial damage occurs in the top left-hand edge of the panel and propagates down that side of the target. The displacement curve flattens around the time of the first indication of damage and peaks at a value of 131 mm, corresponding to a maximum strain of approximately 3%. At 3% strain, the front face sustained inter-laminar skin failure and front-ply fibre breakage, whilst the core suffered cracking from skin to skin, with the rear skin remaining intact. The severity of the failure increased towards the centre of the panel. The 3D DIC results showed the discontinuity in the strain field distribution caused by a separation of skin and core, as well as allowing the shifts in the deformed shape to be observed. From this it can be concluded that 3D DIC can be a powerful tool for monitoring structural integrity and identifying damage mechanisms in even extreme loading cases [38].

LeBlanc and Shukla [39] examined the response of a curved glass fibre reinforced polymer to explosive loading in an underwater environment. The composite plies are composed of e-glass fibres with a [0°/90°] biaxial layup, with a vinyl ester matrix resin. The panels consist of 3 plies, bonded by vacuum infusion, giving a finished part thickness of 1.37 mm and 62% fibre content by mass. The convex face of the panels has a radius of curvature of 182.8 mm, with the curved section of the panel being 228.6 mm in diameter. The curved section of the panel is mounted with the convex surface towards the incoming shock front. LeBlanc and Shukla chose this geometry as it represents common geometries used in underwater applications (such as submarine hulls). The shock wave is generated by a conical shock tube, initiated by an explosive charge. The high-speed imagery was captured by a pair of Photron SA1 cameras (Photron USA, Inc., San Diego, CA, USA) at a frame rate of 20,000 FPS

and the 3D DIC data processed by the VIC-3D software package (Correlated Solutions, Inc., Irmo, SC, USA). One of the main purposes of the experimental component of this study was to obtain data to correlate and validate computational simulations run with a finite element code. The authors did find a high level of correlation between the experimental results and the computational simulations. Their 3D DIC results recorded a peak displacement in the centre of the panel of 80 mm and 55 mm for a point mid-way between the centre and clamped boundary. The peak velocity occurs at the beginning of the event, reaching a magnitude of 20 m/s. The main damage mechanism the authors reported on was delamination between the plies, with minimal fibre rupture or matrix cracking [39].

Kumar and Shukla [40] conducted experiments looking at five different types of glass panels loaded through a shockwave. The types of panels the authors tested were: clear glass, tempered glass, wired glass, sandwiched glass and laminated sandwiched glass with a protective film on both outside faces. Each specimen was 305 mm × 305 mm × 6.5 mm (except the laminated sandwiched glass, which was 1 mm thicker due to the protective film on the front and back sides) and made from soda-lime-silica glass. The shock was imparted to the glass panels via a shock tube, with all samples being subjected to the same level of incident pressure. Kumar and Shukla used a pair of Photron SA1 cameras (Photron USA, Inc., San Diego, CA, USA) running at 20,000 FPS to capture the imagery used in the 3D DIC. The image processing for the 3D DIC was done through use of the VIC-3D software program (Correlated Solutions, Inc., Irmo, SC, USA). The authors show that the laminated glass panel was the only panel to not fail catastrophically, and had the second largest deflection with 1.7% strain. In comparison, the plain glass had a strain of 0.01%, the wired glass, 1%, the tempered glass, 2%, and the sandwich glass with 5% strain before failure. Their results show that the lamination of the sandwich glass significantly improved the blast mitigation properties of the laminate, resulting in delayed deflection and damage propagation to prevent catastrophic failure. The laminated glass panel did experience fragmentation and cracking in the glass, with the protective film containing the glass fragmentation and preventing flying debris. The catastrophic failure of all of the other panels resulted in high-speed fragmentation debris [40].

Arora et al. [41] examined the response of glass-fibre reinforced polymer sandwich panels and laminate tubes to blasts in air and underwater environments. The sandwich panels were made with 2 plies of [0°/90°/±45°] e-glass quadriaxial skins with a styrene acrylonitrile (SAN) foam core (ranging from 30 mm to 40 mm), infused with a Prime-LV epoxy resin. The air blast panels had a size of 1.6 m × 1.3 m and the underwater panels had a size of 0.4 m × 0.3 m. The authors sized the air blast panels to represent full-scale face panels of comparable magnitude to real naval structures and sized the underwater panels to allow for sufficient rigid edge restraint/support during the tests. The air blast was created by 30 kgs of C4 plastic explosive, at a standoff distance of 8 m and 14 m. The 14 m standoff distance was setup to expose the panels to an approximate shock pressure of 200 kPa and take the panels to their elastic limit, while the 8 m standoff distance was chosen to inflict damage to the panels. Notably, 3D DIC was only used for the air blast experiments and the authors used a pair of Photron SA3 cameras (Photron USA, Inc., San Diego, CA, USA) running at 2000 FPS, to obtain the high-speed imagery. The high-speed imagery was processed using the ARAMIS software (GOM GmbH, Braunschweig, Germany) package and the results were verified by use of a laser gauge on the centre of the target. Their results show excellent correlation between the 3D DIC and laser gauge of less than 1% error. For the 14 m standoff, the sample with a 40 mm core deflected by 63 mm (1% major principle strain) and the sample with a 30 mm core deflected by 78 mm (1.25% major principle strain). The 3D DIC did detect signs of mild sub-surface core cracking from early discontinuities in the major principle strain plots. To determine the failure diagnostic capabilities of the 3D DIC technique, a panel with a 40 mm thick core and 8 m standoff distance was used. The authors reported a flattening in the displacement curve near the maximum deflection (131 mm out-of-plane displacement with a 3% strain) that corresponded to crack formation in the skin of the panel. Their results demonstrated that 3D DIC can be a very powerful tool for monitoring the structural integrity of various materials and identifying damage mechanisms occurring even when subjected to extreme loading cases [41].

Gardner et al. [42] investigated sandwich structures with e-glass composite skins, SAN foam cores and polyurea interlayers. The e-glass composite skins had woven fibres laid up in a [0°/45°/90°/−45°] orientation and were 5 mm thick. The SAN foam cores consisted of 3 layers of foam with low, mid-range and high density, with an overall thickness of 38 mm. The first two layers of the core were low and mid-range density and had a thickness of 12.7 mm, with the third being high density and 6.35 mm thick. The authors had two configurations, (i) one located at the polyurea layer just behind the front face sheet, and (ii) the other located it just in front of the back sheet. Gardner et al. exposed the configurations to two loading conditions, an incident peak pressure of 1 MPa and wave speed of approximately 1000 m/s and an incident peak pressure of 1.5 MPa and a wave speed of approximately 1300 m/s. The high-speed imagery was captured with a pair of Photron SA1 cameras (Photron USA, Inc., San Diego, CA, USA) operating at 20,000 FPS and no information was given regarding the software used to process the 3D DIC imagery. The authors show that both configurations exhibited maximum deflection in the low-density foam layer, but configuration (i) had a more uniform load distribution and exhibited maximum compression later in the deformation history. This maximum compression later in the deformation history leads to less transmission of the load to the medium density foam layer. The 3D DIC shows that configuration (i) exhibited a maximum strain of approximately 2.6% across the central region and a maximum out-of-plane velocity of 30 m/s, compared to configuration (ii), which had a maximum strain of approximately 1.625% and a maximum out-of-plane velocity of 24.375 m/s. This shows that configuration (ii) had about 35% less maximum strain and 15% less maximum out-of-plane velocity. Gardner et al. reached the conclusion that the present of the polyurea on the blast-receiving side, configuration (i), amplified the destructive effect of the blast, promoting the failure of the composite sandwich panels. In contrast, placing the polyurea layer on the side opposite the blast-receiving side has a significant positive effect on the panel's response to failure mitigation and energy absorption [42].

Wang and Shukla [43] examined the performance of e-glass vinyl ester composite face sheet/foam core sandwich structures, subjected to compressive in-plane loading prior to shock loading. The skins were e-glass vinyl ester composite with a [0°/45°/90°/−45°] weave and a thickness of 3.8 mm. The foam core had a thickness of 25.4 mm. The authors used three in-plane compressive loads, 0 kN, 15 kN (corresponding to ~20% critical buckling load), and 25 kN (~33% critical buckling load). The blast wave had a peak incident pressure of 1 MPa, with a wave velocity of 1000 m/s. To obtain the high-speed imagery for the 3D DIC, the authors used a pair of Photron SA1 cameras (Photron USA, Inc., San Diego, CA, USA) running at 20,000 FPS. To process the imagery, the authors used the VIC-3D software (Correlated Solutions, Inc., Irmo, SC, USA). The 3D DIC showed that the out-of-plane deflections for the case with no compressive in-plane loading and the 15 kN loading had comparable deflection, with the specimen with 25 kN of in-plane loading exhibiting higher deflections. The authors recorded a reduction in the in-plane stress, which corresponds to cracking in the core material, in the case with no in-plane loading. In comparison, the drop in the in-plane stress that corresponds to core cracking is significantly smaller and occurs significantly later in the cases with compressive in-plane loading. Overall, higher levels of compressive loading cause more damage in the front face sheet, larger out-of-plane deflection and higher in-plane strain on the back face sheet. This leads to the overall blast resistance of the sandwich composite being significantly reduced [43].

Kumar et al. [44] examined panels of 2024 T3 aluminium with curvature exposed to a shockwave from a 1D shock tube. The panels were 2 mm thick and had curvature of (i) infinite (flat plate), (ii) 304.8 mm and (iii) 111.76 mm radius of curvature. The high-speed imagery was captured by a pair of Photron SA1 cameras (Photron USA, Inc., San Diego, CA, USA) running at 20,000 FPS and was processed by the VIC-3D software package (Correlated Solutions, Inc., Irmo, SC, USA). The authors show that the deflection rate was approximately the same in all three panels, but the panels with curvature reached a higher deflection when compared to the flat panel. This was caused by the flat panel stretching more, which limits the deflection. The 3D DIC showed that the initial velocity was the same for all three panels, but the velocity in the flat panel decayed faster when compared to the other two panels.

The out-of-plane acceleration was the highest for the plat panel and decreased for the panels with curvature, with (iii) having the lowest acceleration. The panel with the highest initial curvature, (iii), experienced the least plastic deformation. Given that this panel also had practically no yield line formation (when compared to the other two panels) leads to the conclusion that panel has the best blast mitigation properties [44].

Spranghers et al. [45] examined the response of aluminium test plates to an air blast. The aluminium plates had a thickness of 3 mm and were made from EN AW-1050A H24 grade and was clamped to a steel frame. The explosive used to generate the air blast was a 40 g spherical shape of C4 plastic explosive with a standoff distance of 250 mm. The high-speed 3D DIC imagery was captured with a pair of Photron Fastcam Ultima APX-i2 (Photron USA, Inc., San Diego, CA, USA) running at 6000 FPS. No information was given about the software the authors used to process the high-speed imagery. The 3D DIC showed that the initial out-of-plane displacement is approximating a rectangular shape (due to the plate clamping boundary) and evolves into a sinusoidal shape (due to decay of the blast loading and the deformation only being due to the plate inertia). The blast impulse imparts momentum to the aluminium plate during a very short loading period, imparting enough momentum to continuously deform the plate after termination of the loading from the shock wave. The deformation history consists of two major parts, an initial part consisting of highly plastic behavior reaching a maximum displacement, and a second regime of elastic rebound and elastic vibration with damping. The 3D DIC revealed that in the initial stages, the maximum normal strain occurs close to the boundaries, with strains developing in the centre of the plate after inertial forces have taken over. The in-plane strain is initially very small, eventually increasing in the central region of the plate in a typically bi-axial manner. Initially, the maximum stain rates occur close to the clamped boundaries, with the maximum moving to the centre of the plate later in the blast response [45].

Pickerd [46] conducted a study into the experimental requirements of 3D DIC in a synergistic blast and projectile event scenario. Initially, the authors validated the 3D DIC on an aluminium tensile test, comparing the 3D DIC results to strain gauge data. The authors used a pair of Photron SA5 cameras (Photron USA, Inc., San Diego, CA, USA) for high-speed image capture and the ARAMIS software (GOM GmbH, Braunschweig, Germany) for image processing. After these initial tests, a number of factors to be considered during testing of internal detonations of blast boxes were reported on. To protect the cameras from blast fragmentation, the authors used mirrors with the cameras behind blast protection, as splinter-proof ballistic plexiglass caused the image quality obtained through them to be of low quality. Another area reported on was the consistency of the speckle pattern. The consistency of the pattern can have a significant effect on the accuracy, with areas of singular colour reducing the amount of data obtainable from that location. Once the blast wave impinges on the mirrors, vibration will invalidate any data obtained after this interaction. Lighting is also an important factor reported by Pickerd. The authors noted that ambient outdoor lighting conditions are sufficient, even in overcast conditions. Too much lighting can lead to overexposure, producing inconsistencies in the light levels and leading to data loss in the 3D DIC system. Both small patches of overexposure and changes in light levels from the reference image can cause a loss of data. Lastly, the authors noted that for highly dynamic events, paint adhesion is critical to prevent data loss during the event [46].

Kumar et al. [47] examined the response of curved carbon fibre composite panels to shock wave loading. The panels had a curvature of (i) infinite (flat), (ii) 305 mm, and (iii) 112 mm radius of curvature and are made from 32 layers of carbon fibre with a ([0°/90°/45°/−45°]4s) layup, giving an overall thickness of 2 mm. The shock wave was generated by a shock tube and three shock pressures were generated, ranging from 3 MPa to 8 MPa. A pair of Photron SA1 cameras (Photron USA, Inc., San Diego, CA, USA) operating at 20,000 FPS was used by Kumar et al. to capture the high-speed imagery, and the VIC-3D software package (Correlated Solutions, Inc., Irmo, SC, USA) was used for image processing. The authors show that the failure loading for the panels were (i) 3.65 MPa, (ii) 3 MPa, and (iii) 7.78 MPa. The 3D DIC showed that during the early parts of the loading, the out-of-plane deflection contours were not influenced by the boundary conditions. Later in the loading, the stress

waves deflecting off the boundaries cause a change in the deflection contour shape. The authors used the 3D DIC to extract the out-of-plane deflection, velocities and in-plane strain at the centre point of the three panels. All of the panels exhibited the same surface velocity (35 m/s) for the initial component of the response, though panels (i) and (ii) exhibited higher deflection than panel (iii). It also showed that panels (i) and (ii) showed similar deflection trends until panel (ii) failed through delamination and fibre failure. All three panels showed almost the same in-plane strains at the centre point, pointing towards an isotropic behavior in the three panels. Panel (ii) exhibited higher in-lane shear strain, explaining the catastrophic failure in the panel. At the time that panels (i) and (iii) have a velocity of 0 m/s, panel (ii) retained a velocity of 20 m/s [47].

Abotula et al. [48] conducted a study to investigate the performance of Hastelloy X under a combination of extreme mechanical and thermal environments. This was accomplished by loading the plates with a shockwave, while heating them to high temperatures. The Hastelloy X plates were 3 mm thick and impacted with a peak pressure of 0.25 MPa and a wave speed of 665 m/s. The plates were heated to 360 °C, 700 °C and 900 °C. The high-speed imagery was captured by three Photron SA1 cameras (Photron USA, Inc., San Diego, CA, USA) running at 30,000 FPS. Two of the cameras were used to conduct 3D DIC and a third perpendicular to the side surface of the specimens to capture the side-view deformation images. The authors show that once a material exceeds a certain temperature, self-radiation of the heated material leads to decorrelation effects in the images, as reported by Pan et al. [49] This is due to the radiated light from the heated material dramatically intensifying the brightness, while decreasing the contrast of the captured image [48]. By using an optical band pass filter and a spatial-domain cross-correlation algorithm developed by Pan et al. [50], Abotula et al. were able to obtain images up to 1200 °C [48]. To maintain the speckle pattern, the authors used a commercially available flame proof paint that is able to sustain temperatures up to 1200 °C. The authors show that the maximum impulse imparted to the specimen increases by approximately 41% between 25 °C and 900 °C. In addition, the maximum deflection also increased with temperature (5.2 mm at 25 °C compared to 13.5 mm at 900 °C), but occurs at a later time. Abotula et al. were able to divide the structural response into two stages: the first stage where the effects of the boundary conditions are not dominant, and a second stage where the plate is brought to rest by plastic bending and stretching. As the temperature increases and the maximum deflection increases, the maximum in-plane strain values also increase [48].

Spranghers et al. [51] conducted a study, to show that the use of high-speed 3D DIC and finite element modeling allows for direct identification of the material response, without the need for additional testing. The sample is a 3 mm thick, grade EN AW-1050A H24 aluminium plate. The air blast is created with 40 g of C4 plastic explosive, with a standoff distance of 250 mm. The authors use a pair of Photron Fastcam SA5 high-speed cameras (Photron USA, Inc., San Diego, CA, USA) operating at 25,000 FPS. The authors did not provide any information about what software the authors used to process the high-speed imagery. The free air blast load makes it possible for one to load a plate specimen at different strains and strain rates in different zones. This makes their test suitable for material identification using inverse methods, for which heterogeneous displacement and strain fields are an advantage. Their results show that inverse methods, when combined with 3D DIC, can be successfully used for the determination of the material parameters, as far as complex heterogenous deformation fields are available [51].

Louar et al. [52] conducted an experimental study into the loading of thin square aluminium plates, through an explosively driven shock tube (EDST) and free air blast. Initially, the authors used a 15 mm thick plate to determine the reflected pressure in the EDST, to validate the method of producing a shock loading. The authors show that the EDST was able to produce an improvement in the variability of the incident pressure measurements and impulse, when compared to free air blasts. In addition, the EDST was able to produce significantly higher incident pressure and impulse, than is possible with a free air explosion of the same charge size. After validating the EDST, Louar et al. changed to a 2.5 mm thick aluminium plate and conducted DIC to determine the displacement, strain

and strain rate. The authors used a pair of Photron Fastcam SA5 high-speed cameras (Photron USA, Inc., San Diego, CA, USA) running at 25,000 FPS, to obtain the imagery used in the 3D DIC. The authors validated the DIC measurements with strain gauges and found very good agreement between the two. The authors show that the aluminium plates exposed to free air blast underwent a response that is separated into three stages. The first stage is a compressive through-thickness stress wave that is generated by the interaction between the blast wave and the panel, inducing reaction forces in the fixed boundaries. The second stage is a transfer of momentum and kinetic energy to the panel. The reaction forces generate a stress wave that propagates from the boundaries to the centre of the panel. While propagating, the stress wave produces bending and shear deformations. This causes the panel to move in the z-direction, initially with a square-like shape due to the clamping frame, and proceeding to a circular profile when the maximum deflection point reaches the centre of the plate. The final stage is an elastic vibration and stabilizes in its final shape. Their results show that the loading time for an EDST pressure wave was higher than with the free air blast. There is evidence that the stress/strain propagation patterns are different between the air blast and EDST. The EDST produces a more localized effect, generating a stress/strain wave at the borders of the loaded area and propagating towards the edge and centre of the plate. In comparison, the free air blast causes the stress/strain waves to be generated at the edges and propagate towards the centre. In addition, the EDST provided better reproducibility in terms of distribution, also providing better symmetry, and magnitude of the loading [52].

Del Linz et al. [53] examined the response of annealed glass panels with a polyvinyl butyral (PVB) interlayer to explosive charge in a free air blast. The annealed glass was comprised of two 3 mm thick plies of annealed glass with a 1.52 mm PVB interlayer. The air blast was created by either 15 or 30 kgs of an explosive, with a TNT equivalence of 1.09 and standoff distances of 13, 14 and 16 m. The authors used a pair of Photron S3 cameras (Photron USA, Inc., San Diego, CA, USA) at 1000 FPS to obtain the imagery for the 3D DIC. Del Linz et al. took the blast experimental data to calculate the reactions of both the pre-cracked and post-cracked panels. Due to the different behaviour of the glass in the two situations, the authors employed separate approaches. For the pre-cracked panel reactions, the authors used finite element modelling to estimate the relative magnitude of the bending and membrane stress from the DIC data collected on the back face. The authors show that the trends of the FEA and the experimental deflections were similar in the range of interest. The comparison of measured and estimated failure deflections produced similar results for all experiments considered. In the post-cracked regime, the authors show that the stress–strain curve of the experimental data was very noisy. For two of the panel edges in their first experiment, data uncertainties prevented them from reliably estimating the material coefficients. By calculating the material models off the opposite sides of the windows on those experiments, the authors were able to determine the material properties. Their results show that the majority of the blast energy is reflected back in the direction of the blast origin or vented around the structure, with only 6.8% transferred to the system and 3.1% absorbed by material failure, leading to the conclusion that the absorption capability of the system is relatively small, compared to the energies which are potentially applied to it. The authors show that the post-cracked reaction reached a distinct plateau as the central deflections increased. Del Linz et al. postulate that this could be caused by delamination between the PVB interlayer and the outer glass plies [53].

Pickerd et al. [54] examined the response of steel containers to an internal cylindrical explosive charge. The steel containers had dimensions of 1 m × 1 m × 1 m, with a wall thickness of 5 mm. The containers had two variants, one with a flange that can be bolted to a solid base and a fully enclosed variant. The flanged container had one open end and a 10 mm thick flange, which the authors bolted to a non-responding steel plate secured to a concrete block. The flange was welded to the side of the 5 mm thick container wall using a 10 mm thick reinforcing square section, which was welded on the outside corner of the container in the interface between the container and flange. The fully enclosed containers were not bolted down, but instead suspended on wooden blocks. The explosive was uncased PE4 plastic explosive with a length/diameter ratio of 1 and a mass of between 245 g to

431 g. The DIC imagery was taken at 7000 FPS, but the authors did not specify the specific model of camera used. The authors used the ARAMIS software (GOM GmbH, Braunschweig, Germany) package to process the high-speed imagery. Most of the containers experienced failure around the welded joint from weld tear initiation observed during the initial expansion of the container. In the majority of experiments the authors reported on, the container would reach maximum deformation, then contract, and during a second expansion, the initial weld tear would propagate along the welded joint, resulting in the catastrophic failure of the container. In the experiments with the largest charge mass, their results show the initial weld tear and subsequent weld tear propagation occurred prior to the container reaching maximum deformation. Failure in those cases occurred due to the applied load significantly overmatching the strength of the welded container. Pickerd et al. reported on the strain measurements recorded using the DIC. The authors show that the strain profiles were consistent for all events, with only the maximum strain values differing. The authors show that the highest strains were observed at the edges of the container in the welded joints for all events. The second highest regions of strain were located on the face of the container. The region of highest deflection, the centre of the face, recorded the lowest strain. The strain rate was approximately an order of magnitude greater on the weld joint to the face of the container [54].

Chen et al. [55] examined the response of thin circular plates to confined explosions. The circular plates were made from ASTM C11000 copper, h62 brass or $\alpha$-titanium alloy (Ti 50A). All of the plates were tested with a 2 mm thickness, with the copper also being tested with 3 mm thickness. The explosive was a desensitised RDX plastic explosive, with a mass ranging from 3 to 4 g and a diameter of 20 mm. The explosion was confined in a small-scale stainless-steel cylinder with the test plates bolted onto one end and the other end being welded closed. The authors used a pair of Photron Fastcam SA5 cameras (Photron USA, Inc., San Diego, CA, USA) running at 50,000 FPS to capture the high-speed imagery. The images were processed with the VIC-3D software (Correlated Solutions, Inc., Irmo, SC, USA). The authors show that the plates deformed with the out-of-plane motion centred on the centre of the plate, forming an approximately symmetric dome shape. The out-of-plane velocity begins in the centre of the plate and remains localized around the plate centre, reaching a maximum while the plate deflection is still low. The velocity drops to zero at the same time as the maximum deflection occurs. The principle strain first occurs in the central area of the plate, with a dome shaped strain localization zone appearing in the central area and spreading rapidly outwards towards the boundary region. From this time, another concentric strain zone forms near the boundary region and spreads rapidly inwards towards the plate centre. The highest strains are reported in the area close to the clamped region. The authors reported that the plates only experienced inelastic deformation and no tearing or shearing. In addition, the authors show that the titanium alloy exhibited significantly smaller deformations than the copper or brass specimens, implying that the Ti 50A titanium alloy exhibited far better resistance to dynamic deformation.

Aune et al. [56] examined the response of thin aluminium and steel plates subjected to airblast loading. The aluminium and steel plates had a thickness of 0.8 mm and were clamped to a rigid frame, to attempt to create fixed boundary conditions. The steel plates were made from Docol 600DL, a medium strength, high-hardening and cold-rolled sheet, and the aluminium plates made from EN AW 1050A-H14 alloy, a low-strength, strain-hardened and cold rolled sheet. The air blast was created with a spherical mass of C4 plastic explosive with a mass of 30 g and diameter of approximately 34.5 mm. The standoff distances were 0.125 m, 0.250 m, and 0.375 m for the steel plate and 0.250 m, 0.375 m, 0.50 m and 0.625 m for the aluminium plate. Aune et al. used a pair of Phantom v1610 high-speed cameras (Vision Research Inc., Wayne, NJ, USA) running at 21,000 FPS to capture the imagery for the 3D DIC and an in-house code to process the imagery to obtain deflection, strain and strain rate information. The authors define three types of dynamic elasto-plastic structural response. Type 1 is where the structure oscillates on both sides of its original configuration with a positive (away from the blast origin) final deflection. Type 2 is characterized by more intense pulses, causing oscillation only on the positive side of the original configuration. Finally, type 3 is where the structure may first deform in

the positive direction and continue to rebound to the negative side of the original configuration. Type 3 is very rare and extremely sensitive to the structural and loading parameters. All of the steel plates experienced severe plastic deformation, due to the blast load and the fact that the elastic rebound became smaller as the load-intensity increased. The steel specimens with the 0.125 m and 0.250 m standoff both exhibited type 2 structural response. The steel specimens with the 0.375 m experienced reverse snap buckling during the elastic rebound, oscillating in the opposite direction to the direction of travel of the incident wave. In comparison, the aluminium panels at the closest standoff distance experienced complete failure of the plate boundary, resulting in the plate being torn out of the clamping frame. Aune et al. showed that a transition between large inelastic deformation and complete tearing at the plate boundary occurs at a standoff distance of 0.375 m. The largest standoff distances of 0.50 m and 0.625 m resulted in large inelastic deformation (type 1) and reversed snap buckling (type 3) [56].

Dear et al. [57] performed a study examining the response of glass fibre reinforced polymer (GFRP) and foam sandwich structures to explosive air blasts. The face sheets of the sandwich structures were 2mm thick GFRP and the cores were 40 mm thick foam made from styrene acrylonitrile (SAN), polyvinylchloride (PVC) or polymethaacrylimide (PMI). The air blast was created with a 100 kg TNT equivalent charge, with a standoff distance of 15 m. Dear et al. used a pair of Photron SA1.1 cameras (Photron USA, Inc., San Diego, CA, USA) operating at 5400 FPS to capture the high-speed imagery. The authors used the ARAMIS software (GOM GmbH, Braunschweig, Germany) package to process the 3D DIC data from the imagery. The authors show that the initial displacement and pull-out of the PVC and PMI panels were greater than the panel with SAN foam core. This indicates that the SAN foam core is more resistant to the momentum of the blast and suffers less core damage. Their results showed that around 75% of the PMI panel was damaged, with a mix of debonding and cracking. In comparison, the SAN panel suffered the least damage. In addition, the authors show that there is a trade-off between reduced panel deflection and damage [57].

Ye et al. [58] used 3D DIC to examine the response of metal face sandwich plates to impact by a metallic foam projectile to simulate blast loading. The sandwich plates the authors used were made from PVC foam layers, with densities of 80 $kg/m^3$, 160 $kg/m^3$ and 250 $kg/m^3$, and 5A06 aluminium alloy face sheets. The authors created four configurations for the testing, with all of the configurations having 0.5 mm thick face sheets. The four configurations the authors reported on were: (I) a single layer of 250 $kg/m^3$ foam, (II) 160 $kg/m^3$ foam followed by 80 $kg/m^3$ foam, (III) 80 $kg/m^3$ foam followed by 160 $kg/m^3$ foam, and (IV) three layers of 80 $kg/m^3$ foam. To simulate a blast impact, the authors used a single stage gas gun to propel a metallic foam projectile, 30 mm in length, 40 mm diameter and a density of 350 $kg/m^3$, at the sandwich panels. The authors used both 3D DIC and a digimatic micrometer system to measure the permanent deflection. To capture the high-speed imagery, Ye et al. used a pair of Photron Fastcam SA-Z cameras (Photron USA, Inc., San Diego, CA, USA) operating at 36,000 FPS to capture the images and the ARAMIS software (GOM GmbH, Braunschweig, Germany) to process the images for the 3D DIC. The results from the digimatic micrometer and the 3D DIC are in close agreement. The authors show that there were three stages of dynamic deformation: during the first stage the impact impulse transfers to the front sheet, giving it a velocity; while the core and rear sheet remain static, the second stage is where the core becomes compressed, as the impulse transmits through the core to the rear sheet, giving the whole plate a common velocity and the final stage, where the energy is dissipated and absorbed through bending and stretching. The 3D DIC shows that the maximum strain occurs at the centre of the rear sheet, indicating that tension-tearing deformation in the middle of that face sheet will be one of the major failure modes when the impulsive intensity increases. Their results demonstrate that low-density, but thicker, cores perform the best in reducing permanent deformation of the structure. In contrast, higher density, but lower thickness, cores perform best in protecting the structure from destruction. With mixed core densities, the descending order of core densities has an advantage in blast resistance [58].

Curry and Langdon [59] examined the response of steel plates to close proximity air blasts. The test plates were made from 3 mm thick Domex 355MC; a high strength, hot rolled, low alloy

steel. The air blasts were created with a PE4 plastic explosive charge of masses between 10 g and 50 g and a standoff distance of 40 mm and 50 mm. The 3D DIC imagery was captured with a pair of IDT Vision NR4 S3 high-speed cameras (IDT Vision, Pasadena, CA, USA) and the DIC processing conducted with the Dantec Dynamics Istra 4D DIC software package (Dantec Dynamics, Skovlunde, Denmark). The authors reported that all of the plates exhibited plastic deformation, without any indication of tearing or shearing failure. The authors used the 3D DIC to examine the evolution of the transient out-of-plane deformation profile of the plate and the transient deformation time history of the midpoint of the plate. The initial deformation appears to be localized to a plastic hinge in the centre of the plate. As time progresses, the plastic hinge moves outward to the clamp frame, with a conical section between the midpoint and the plastic hinge. The next stage is the plate forming into a general bell shape with a point of inflection around 100 mm diameter. By the time the plates have reached their maximum deflection, the point of inflection has become less significant and is almost undetectable by the time the plate reaches its final shape. The displacement histories produce very similar profiles at each charge mass. An important observation is that during the transient deflection, the profile does not match the shape of the final deformed shape at any point. The strain measured by the 3D DIC shows that the strain is localized to the centre of the plate, peaking at the centre of the plate and decaying to a very small value approximately $1/6^{th}$ the plate diameter from the centre. The minimum strain coincides with the point of inflection in the permanent deflection profile. The maximum strain increases with an increase in charge mass and decrease in standoff distance [59].

Shukla et al. [60] examined displacement measurements of submerged implodable volumes. Implodable volumes are defined as any structural shell or body that is acted on by an external pressure and contains internal gas at a lower pressure (or vacuum). As a submerged body implodes the acquired momentum of the in-rushing water causes it to over-compress against the structure and produces strong outwardly radiating shockwaves. Shukla et al. examined three cases, a free-field implosion caused by increasing pressure, a semi-confined implosion caused by increasing pressure or explosive initiation, and a closed, confined tube implosion initiated by explosive. The authors used a pair of Photron SA1 cameras (Photron USA, Inc., San Diego, CA, USA) operating at 30,000 FPS to obtain the 3D DIC imagery. The deformations, strain and strain rate were calculated from the imagery through the VIC-3D version 7 (Correlated Solutions, Inc., Irmo, SC, USA). The accuracy of the 3D DIC in an underwater environment was calibrated in a custom designed tank. Shukla et al. showed results from a seamless tube of aluminium 6061-T6 grade, with ends sealed by hemispherical caps. The authors show that the deformation of the specimen begins at the centre and symmetric about the mid-plane. The 3D DIC shows that the centre of the specimen reaches a maximum wall velocity of ~25 m/s and leads to a substantial rate of change in the volume of the specimen. The contact of the two opposite walls moves towards each other with a relative velocity of ~50 m/s and the impact between the two solid surfaces leads to strong sound waves, manifesting as a short duration shock pulse in the water. Their results show that this contact propagates towards the direction of the end-caps with high velocity (~100–200 m/s). In comparison, the case where the implodable volume is contained within a semi-confined tube, the walls only reached a maximum wall velocity of ~12 m/s and a collapse front velocity of ~70 m/s. This decrease is due to the limited amount of potential energy in the limited amount of fluid around the implodable volume. In the case where the implosion was initiated by an explosive, it caused cavitation along the confinement walls, causing various bubble cycles. The high pressure from the first bubble collapse supplies sufficient energy to the specimen to make it unstable and collapse. As there is less potential energy in the surrounding fluid, the walls start with a collapse velocity of ~12 m/s, but slow to 8 m/s before contact with the opposite wall. Shukla et al. was able to use the 3D DIC to capture longitudinally travelling axisymmetric vibrations in the cylindrical shell, which are excited by the local radial compression [60].

Mehreganian et al. [61] examined the response of mild steel and armour steel plates to air blast explosive loading and used 3D DIC to compare experimental results to computational models. The plates had a nominal thickness of 4.6 mm. The air blast was generated by circular disks of PE4

plastic explosive, with a mass ranging from 24 g to 70 g and a standoff distance of 25 mm, 38 mm and 50 mm. The 3D DIC imagery was taken with a pair of IDT Vision NR4 S3 high-speed cameras (IDT Vision, Pasadena, CA, USA) running at 30,000 FPS. The authors used the Dantec Dynamics Istra 4D DIC software package (Dantec Dynamics, Skovlunde, Denmark) to process the imagery. The authors obtained the deflection, strain and strain rate and compared it to the computational models. The authors show that the models varied from between 4.97% and 33.41% for the final deflection of the mild steel plates from the 3D DIC data. The difference between the experimental and computational varied from 2.95% to 30.27% for the ARMOX370T and 7.46% to 26.63% for the ARMOX440T. Their results show that the ARMOX steel panels exhibited lower maximum and residual plastic deformation, due to the higher ability of energy dissipation from the blast, which subsequently increases as the material yield strength increases [61].

Dai et al. [62] studied the response of copper plates subjected to underwater impulsive loading. The plates were made from C11000 copper and had thicknesses of 1 mm, 2 mm and 3 mm. The authors also examined the response when cross and ring pre-notches were cut into the plates. The depth of the notches was 0.5 mm. The pressure pulse was generated by a modified gas gun apparatus. The authors used a pair of Photron Fastcam SA5 cameras (Photron USA, Inc., San Diego, CA, USA) to obtain the high-speed imagery and used the VIC-3D software package (Correlated Solutions, Inc., Irmo, SC, USA) to generate the 3D DIC data. The authors show that the relative out-of-plane error measurements are between 0.84% and 6.83%. For the specimens with no pre-notch, their results show a linear relationship between normalized final deflection and dimensionless impulse. Without a pre-notch, the primary failure mode was large ductile deformation, while the pre-notched specimens exhibited four other typical failure modes. These other typical failure modes are large ductile deformation with local necking, splitting, splitting and tearing and fragmentation [62].

## 4. Where Next?

One of the areas where significant research is being conducted is the enhancement of X-ray DIC techniques to allow for high and ultra-high-speed 3D DIC. Olbinado et al. [63] was able to achieve up to 1.9 million FPS X-ray phast-contrast imaging (XPCI) for the visualization of crack propagation in glass and shock wave propagation in water. From their techniques, a high spatial resolution (effective pixel size of 8 μm) and large field of view (12.8 mm × 8 mm) was achieved at MFPS frame rates. While this study was not DIC, the techniques the authors describe could be extended to perform 3D-DIC in dynamic deformation events [63].

Olbinado et al. [64] extended their previous research by examining three methods of obtaining high and ultra-high-speed synchrotron X-ray imaging, shown in a simplified manner in Figure 4. The method the authors examine is using a frame transfer CMOS camera with in-pixel storage, to achieve frame rates above 1 million FPS without reducing the number of pixels in the image frame. The authors demonstrate this method by showing the wave propagation and attenuation in a laser-shocked epoxy foam. The second method is using a high efficiency X-ray image intensifier combined with a CMOS camera. The experimental the authors used to demonstrate this method was to examine the stress (via XPCI and diffraction imaging) in silicon wafers with heat-induced cracks. Due to the limited acceptance of X-rays by crystalline materials, the diffracted signals used in this method are orders of magnitude weaker than the incident polychromatic X-ray beam, leading to the use of an image intensifier. The authors were able to use a conventional CMOS camera to give thousands of frames needed to follow fracture and failure, as well as corresponding strain dynamics. The third method the authors describe is the use of high resolution sCMOS camera with a gate-able visible light image intensifier. This method is a more sensitive form of XPCI and is achieved by using X-ray optics that directly measure beam refraction in grating interferometry and tracking near-field speckles introduced by randomly structured material. This allowed the authors to obtain an effective pixel size of 1.625 μm with a time resolution of 70.4 μs [64]. This third method the authors describe is similar

to DIC and hence could be adapted to provide high and ultra-high-speed 3D DIC, with very small spatial resolution.

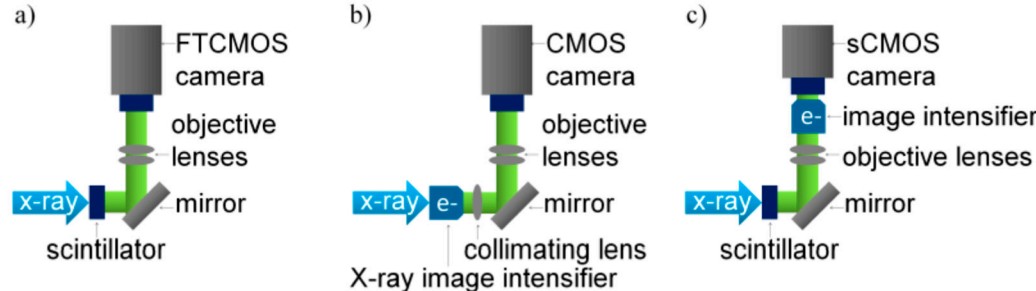

**Figure 4.** Reproduction of Figure 1 of Olbinado et al. [64], showing simplified schematics of X-ray imaging techniques. (**a**) MFPS frame rate, propagation based XPCI; (**b**) kFPS frame rate diffraction topography; (**c**) single-bunch, single grating interferometry and speck based XPCI at ID19. © IOP Publishing Ltd and Sissa Medialab. Reproduced with permission of IOP Publishing. All rights reserved.

Further work has been done with X-ray imagining at the European Synchrotron Radiation Facility by Escauriza et al. [65] The authors were able to achieve an effective frame rate of 5.68 million FPS, by using three Shimadzu Hyper Vision HPV-X2 cameras (Shimadzu, Kyoto, Japan). The authors converted the X-rays to photons, detectable by the cameras with in-line scintillators. The high frame rate was achieved by using the cameras to capture sequential sets of X-ray photons, allowing for all of the pulses created by the synchrotron to be captured. The images from these sets were then stitched together to give a continuous set of images with an interframe time of 176 ns. Their system was able to capture a pixel size of 32 μm and given the main features of the object were significantly larger than the characteristic length, the edge contrast in the recorded images was enhanced, while preserving the attenuation projection of the target. The experiment the authors used to test the method was the impact of a gas gun projectile into an additively manufactured aluminium lattice [65].

Other recent work by Dong and Pan [66] has looked into using a UV 3D DIC system to measure the surface of an ablating body in an arc-jet facility. The authors used two UV cameras to record images of the self-luminescent surface of an ablating body, shown in Figure 5. There are two key challenges to performing 3D DIC in high temperature environments, imaging and speckling. The first challenge is due to the images from regular 3D DIC systems being seriously degraded by intense thermal radiation, emanating from both the test specimen and the heating device. The authors overcame this challenge by using an active UV illumination and bandpass filter imaging. This combination offered extraordinary performance in acquiring high-quality surface images with almost constant image contrast in extreme environments. The speckling challenge was overcome by using the sample surface as the speckle pattern. As the natural texture on the surface changes during the ablation, the authors used an incremental calculation strategy using updated reference image pairs for each 3D DIC analysis. The authors validated the viability and efficacy of the new method, by conducting ablation tests in an arc-heated wind tunnel. The experimental results clearly show the ablation at the front of the specimen, with a sufficient and almost constant image contrast. These results also confirm the capability of the UV imaging system in suppressing the adverse influence of thermal radiation [66].

This section shows that of the potential paths that future 3D DIC can be extended into for high energy deformation events, the most important one is the use of non-visible light. The use of non-visible light allows for high-speed 3D DIC to be performed on internal structures, for example the internal structures of foams, with high spatial resolution, by using X-rays, or extreme thermal environments, with the use of UV light.

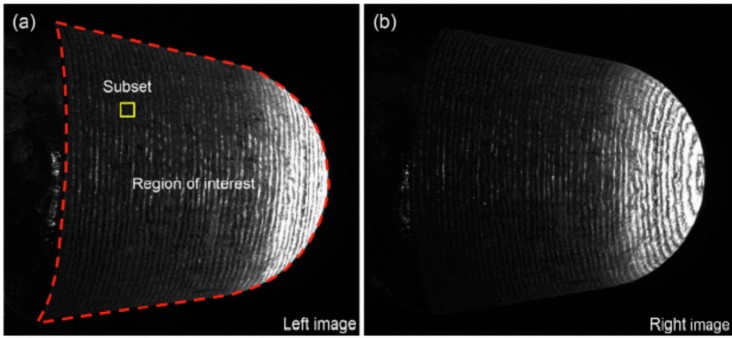

**Figure 5.** Example of UV illuminated images captured by Dong and Pan [66] at 1000 °C: (**a**) this is the image captured by the left camera at 1000°C, showing the region of interest and the subset used by the DIC software; (**b**) this is the image captured by the right camera at 1000°C. Reproduced from Dong and Pan [66], with permission from Elsevier.

## 5. Conclusions

In conclusion, 3D DIC is a relatively new technique, that is quickly becoming a very valuable tool as a non-invasive method of measuring three-dimensional deflection, velocity, acceleration, strain and strain rate in materials. With several companies producing commercial software packages for the processing of sets of high-speed imagery, it is becoming even easier to implement and use 3D DIC for the characterization of material response in both ballistic and blast events. This technique has even been extended into the X-ray spectrum, allowing one to measure the internal deformation and strain. The following is a list of the observed benefits of using DIC in dynamic experiments:

1.  It is non-invasive and therefore a preferred method when 'in-material' gauges could affect the result of the test.
2.  It provides full field measurements of strain and deflection, whereas 'in-material' gauges only provide point measurements.
3.  By utilizing different spectrums of non-visible light, 3D DIC can be performed in regimes where heat or particulate matter may obscure or distort the image (X-ray imaging), or regions where an 'in-material' gauge would be destroyed (UV imaging), for example high temperature applications.
4.  By utilizing X-ray imaging, 3D DIC can be performed on internal surfaces or inside of objects where it would be very difficult or impossible to wire up 'in-material' gauges without changing the structure.

As in so many areas of engineering and science, our understanding of the behaviour of armour materials under impact loading conditions has been enhanced by analytical models and finite element simulations, coupled with laboratory testing techniques that probe the high strain-rate response. In particular, the DIC technique has demonstrated that it is an invaluable approach to couple with finite element techniques to provide hi-fidelity data, to validate numerical simulations.

**Author Contributions:** Conceptualisation, C.L.E.; review and summarising of papers, C.L.E.; writing—original draft preparation, C.L.E.; writing—review and editing, C.L.E., P.H. All authors have read and agreed to the published version of the manuscript.

**Funding:** This research has been funded by Integrated Personnel Protection STC, Land Division, Defence Science and Technology Group, Australia, research agreement ID9028.

**Conflicts of Interest:** The authors declare no conflict of interest.

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
