# Peer review of "Visual Methods to Assess Strain Fields in Armour Materials Subjected to Dynamic Deformation—A Review"

_applsci, doi:10.3390/app10082644_

Round 1
Reviewer 1 Report
The current review paper is written well and explains in depth about 3D-DIC analysis for armor applications. The detailed analysis of each paper and its relevance to 3D DIC and accuracy of dynamic deformation variables such as deflection, velocity, acceleration, strain field and strain rates are discussed with examples. The structure of the paper is well organized. In ballistic application, the accuracy of results depends on many factors such as high speed cameras, diffused lights, reliable speckle pattern, post processing of images in DIC software etc. All these parameters are explained using appropriate citations. However, it would be nice to add a small generalized discussion on strain fields captured through 3D DIC related to localized deformation - kink and shear bands in polymer matrix composites, when loaded in compression and bending. Upon a shock, there may be multiple kink bands develop in the material and propagate through material thickness that could induce catastrophic failure and increase chances of brain injuries and cardiac arrest in soldiers. Measuring the full field strain field in unstable kink band growth (dynamic event) is quite challenging considering the length scale of the width of kink band. This may require a larger field of view (FOV), increased depth of field and reliable speckle pattern that may not shear during large deformation. There are many relevant papers in existing literature, but information on such failure mechanisms should be reflected. I suggest to read following paper hyperlinked as "Suggested Reading" that explains full field strain measurements in kink band using 2D-DIC, but a few specifics could be related to 3D-DIC in particular a reliable speckle pattern. In addition, you can explore relevant papers in literature.
Suggested Reading
Author Response
1 - A paragraph has been added to the first section of the paper "The Technique and Origin of Digital Image Correlation (DIC)" discussing the challenges of using DIC to resolve the strain of small scale localised deformations as well as global strains. This paragraph references the paper in the suggested reading as an example of this particular challenge. The challenge in creating a speckle pattern of sufficient size to resolve small scale local deformation is also discussed.
Reviewer 2 Report
The review presented is wide and well structured. It is a well-explained paper, reaching an easy reading and understanding of it. However, I would highlight the following correction to consider:
1-Along the text: "They described...", "They found..." under my point of view these expressions can be changed by other more scientific expressions as, for example: presented in the paper... the work shows... The authors show... etc. More scientific sentences.
Author Response
1 - The writing style has been edited to make the wording more appropriate for a scientific document. Numerous changes have been made throughout the document. An example of this is replacing "they" with "the authors".
Reviewer 3 Report
There is good and comprehensive review. It presents a very useful 3D Image Correlation (3D DIC) tool predicted to measure of parameters of dynamic high-strain rate response of ballistic protection materials during high-speed impacts.
The paper is written well.
Author Response
This reviewer did not request any changes.